# Three Pathways of Cancer Cachexia: Inflammation, Changes in Adipose Tissue and Loss of Muscle Mass—The Role of miRNAs

**DOI:** 10.3390/jpm12091438

**Published:** 2022-08-31

**Authors:** Iwona Homa-Mlak, Dominika Pigoń-Zając, Paweł Wawrejko, Teresa Małecka-Massalska, Radosław Mlak

**Affiliations:** 1Department of Human Physiology, Medical University of Lublin, Radziwiłłowska 11, 20-080 Lublin, Poland; 2Department of Nursing Education, Johns Hopkins University School of Nursing, Wolfe Street, Baltimore, MD 21205, USA

**Keywords:** miRNA, cancer cachexia, inflammation, browning of white adipose tissue, muscle atrophy

## Abstract

According to the World Health Organization, in 2018, cancers, along with over 18 million new cases and over 9.5 million deaths remained one of the main causes of mortality globally. Cancer-cachexia, also called wasting syndrome is a complex, multifactorial disorder characterized by progressive skeletal muscle mass loss, with or without adipose tissue atrophy. It is considered as a state of cancer-related malnutrition (CRM) accompanied by inflammation, that is irreversible despite the introduction of nutritional support. Indication of markers of pre-cachectic state seems to be urgently needed. Moreover, such markers have also potential to be used in the assessment of the effects of anti-cachexia treatment, and prognosis. miRNAs are non-coding RNA molecules that are about 20–30 nucleotides long. Single miRNA has the potential to control from few dozen to several hundred different genes. Despite the fact, that the number of miRNAs keep growing. we are making steady progress in establishing regulatory targets and their physiological levels. In this review we described the current knowledge on the impact of miRNAs on processes involved in cancer cachexia development: inflammation, adipose tissue remodelling, and loss of muscle mass both in animal models and the human cohorts. The available studies suggest that miRNAs, due to their properties, e.g., the possibility of regulating even hundreds of different genes, signalling pathways, and biological processes by one molecule, but also due their stability in biological material, the fact, that the change in their level reflects the disease status or the response to the applied treatment, they have great potential to be used as valuable biomarkers in the diagnosis, treatment, and prognosis of cancer cachexia.

## 1. Introduction

### 1.1. Cancer Cachexia

According to the World Health Organization, in 2018, cancers, along with over 18 million new cases and over 9.5 million deaths remained one of the main causes of mortality globally [1]. Cancer-cachexia, also called wasting syndrome is a complex, multifactorial disorder characterized by progressive skeletal muscle mass loss, with or without adipose tissue atrophy. It is considered as a state of cancer-related malnutrition (CRM) accompanied by inflammation, that is irreversible despite the introduction of nutritional support [2,3]. Most cancer patients are burdened by cachexia (in advanced stages of the disease, it is noted even in 80%of cases). What is more, it is considered a negative prognostic factor regardless of the tumor site or stage [4]. Cancer cachexia is the major cause of death for 22–30% of cancer patients. Moreover, due to the constantly increasing incidence of cancer, it is also expected, that the clinical importance of this syndrome will further rise in the future [4,5]. Despite its serious implications, wasting syndrome is commonly omitted or misdiagnosed, especially at the beginning of the treatment. Diagnosis of this syndrome requires an examination of weight loss (5% of body weight loss over a 6-month period), body mass index (BMI; lower than 20 kg/m^2^), and skeletal muscle status (low muscle mass—sarcopenia) [3,6]. Interestingly, cancer cachexia development seems to be depend not only on the tumor type, site and stage, but also on individual predispositions (genetic and epigenetic alterations, initial body composition and BMI, food intake, gut microbiota, physical activity and comorbidities) [4,7,8]. Despite numerous studies in this area, the etiopathogenesis and molecular pathways of this syndrome remain poorly understood. It is believed, that cancer cachexia is primarily caused by abnormal metabolism, anorexia, and inflammation. It appears, that exactly chronic inflammation triggered by both tumor- and host-derived factors may be a key initiator of this syndrome [7]. Muscle atrophy is caused by an imbalance in muscle protein synthesis and degradation, that results in a decrease in myofibrillar and sarcoplasmic proteins, as evidenced by muscle fiber shrinkage. Nevertheless, the nature of the critical factors causing muscle atrophy in cancer cachexia remains unclear [9]. However, it should be noted, that weight loss in cancer patients can not only be due to muscle mass loss but also adipose tissue loss [10]. It has been demonstrated, that in cancer patients, fat loss is primarily caused by lipolysis rather than irreversible adipose cell degeneration caused by apoptosis, with an overall increase in lipolysis of approximately 50% [11]. It has also been proposed, that the browning of white adipose tissue (WAT), as well as vicious biochemical processes, that occur in brown adipose tissue (BAT) during which oxidative phosphorylation is not linked to ATP synthesis and leads only to heat production, resulting in increased and ineffective energy consumption contribute to the development of cachexia. Many explanations have been proposed for tumor-induced lipolysis. Despite the presence of inflammatory cytokines released by infiltrating macrophages, activation of triglyceride lipase in fat, and loss of activated protein kinase of 5’ AMP, the mechanism by which fat loss contributes to cancer cachexia remains unknown [2]. Since cancer cachexia is considered irreversible once relevant weight loss occurs, using weight loss as a diagnostic criterion appears to be clinically insufficient [12]. Therefore, indication of markers of pre-cachectic state seems to be urgently needed. Moreover, such markers have also potential to be used in the assessment of the effects of anti-cachexia treatment, and prognosis.

### 1.2. miRNA

miRNAs are non-coding RNA molecules that are about 20–30 nucleotides long [13,14]. Disease development can result from altered miRNA expression, that interferes with and disturbs physiological systems [13,15]. The cell cycle, cell division, programmed cell death, neoplastic transformation, and metastasis can all be regulated by miRNAs [16,17]. The role of miRNAs was studied in the etiopathogenesis of different disease entities including cardiovascular disorders [18], neurological illnesses [19], kidney diseases [20] cancers of the lungs [21], head and neck [22], breasts [23], and prostate [24]. Most studies focus on the assessment of miRNAs as potential tumor biomarkers, that are useful in diagnosis and prognosis. Additionally, it is believed, that miRNAs may constitute potential therapeutic targets in cancer patients [25]. Cancer patients may benefit from treatments that mix complementary miRNAs. There are numerous clinical trials (in vitro, in vivo, phase I, and phase II) on the use of miRNA-based therapeutics in cancers such as myeloma (miR-1258 as a target) [26], colorectal cancer (miR-145 as a target) [27], NSCLC (miR-34a as a target) [28] and chronic lymphocytic leukemia (CLL) (miR-155 as a target) [17,29]. It should be highlighted, that a single miRNA has the potential to control from few dozen to several hundred different genes. Despite the fact, that the number of miRNAs keep growing, we are making steady progress in establishing regulatory targets and their physiological levels [16,30].

#### 1.2.1. The Role of miRNAs in the Regulation of Proinflammatory Cytokines Involved in Cancer Cachexia

Tumor Necrosis Factor-α (TNF-α), gamma interferon (IFN-ɣ), interleukin-1 (Il-1), and interleukin-6 (Il-6) are examples of pro-inflammatory cytokines that may have an impact on so-called “central” and “peripheral” metabolic pathways and on their functionality. The influence of the proinflammatory cytokines on the central pathways may lead to changes in the hypothalamus namely by altering the functioning of the hunger and satiety centers, that are involved in the regulation of food intake. In this case, the anorexigenic pathway (pro-opiomelanocortin (POMC) and cocaine- and amphetamine-regulated transcript (CART)) is stimulated, while the orexigenic pathway (Ghrelin-Neuropeptide Y/Agouti-related protein) is inhibited, promoting catabolism and anorectic processes. The peripheral pathway includes protein degradation, lipolysis, and insulin resistance. This processes are regulated by TDF (tumor-derived factor) in the form of PIF (Proteolysis Inducing Factor) and LMF (Lipid Mobilizing Factor) [31]. Inflammation promotes fatigue, reduces physical activity, and causes anorexia and weight loss. It can impair the function of the muscle tissue even in the case of proper energy supply. In cancer patients, it is associated with increased toxicity of treatment (which in turn may lead to a change in therapy or even its discontinuation), deterioration of the quality of life, and poor prognosis. Continued weight loss impairs exercise capacity all the more and worsens the clinical condition of the patient [32].

In a cohort of 70 patients with head and neck cancer (HNC) (without cachexia, *n* = 33, with cachexia, *n* = 37), Powrózek et al. examined the role of miR-130a level in predicting cachexia. Authors demonstrated a link between reduced miR-130a expression and a higher risk of weight loss. In 33 cases, expression was elevated, whereas in 37 individuals, expression was decreased. Low expression was linked to higher plasma TNF-α levels [33]. Fabbri et al. examined the level of several miRNAs in a B6 mouse model of lung cancer (*n* = 18) and cell cultures (A549, SK-MES, HEK-293). Authors demonstrated that lung cancer cells release miR-21 and miR29a, which bind as ligands to immune cells’ Toll-Like Receptors (TLRs) (murine TLR7, human TLR8) and trigger an inflammatory response. Tumor progression could result from this mechanism. Fabbri et al. also observed increased secretion of TNF-α and IL-6 by activating the NF-KB pathway [34]. Another study of the relationship between miR-155 and inflammation in cancer cachexia was conducted by Yehia et al. Authors assessed the expression of miR-155 in a group of 203 patients with pancreatic cancer or non-small cell lung cancer (NSCLC). The cohort was divided into a control group: without cachexia (*n* = 94) and a group with cachexia (*n* = 109). Overexpression of miR-155 was noted in the group of patients with cachexia and contributed to the inhibition of negative feedback loops of SOCS1 as well as Foxp3 and TAB2. Researchers have shown that miR-155 is a regulator of the gene encoding TNF-α with a simultaneous pro-inflammatory effect [35] (Table 1). Based on the above in vitro and in vivo studies, it can be concluded that miRNAs can affect signaling pathways and modulate the immune response. Inflammation is observed in patients with cancer cachexia.

#### 1.2.2. Role of miRNAs in Cancer Cachexia-Related Adipose Tissue Remodelling

Adipogenesis is one of the main factors influencing the mass of adipose tissue. This process involves the differentiation and maturation of adipocytes [36,37]. Adipocyte number depends on the level of differentiation of pluripotent stem cells into preadipocytes. In turn, Adipocyte size depends on the extent of preadipocyte differentiation and triglyceride accumulation. Differentiation is tightly controlled by the activation of appropriate signalling pathways [36]. Understanding the molecular mechanisms involved in adipocyte differentiation and maturation and finding biomarkers for these processes may contribute to the indication of new therapeutic targets. It seems that microRNAs may play an important role in adipokinesis by influencing the regulation of transcription factors and signalling pathways [38].

One of the causes of cancer cachexia is an increase in the metabolic rate and their general deregulation [36,39]. In a cancer mouse model, activation of thermogenesis associated with the presence of BAT was observed. The authors suggested that increased metabolism contributes to cachexia development. [40]. WAT is characterized by energy accumulation, while BAT dissipates this energy in the form of heat. It has been observed that the browning of WAT is associated with cachexia and may be caused by lipolysis [4,36,41]. In a cancer mouse model in BAT, a decreased activity of lipoprotein lipase was observed [42]. Patients with cancer cachexia demonstrate increased expression of mRNA for lipase. Moreover, in patients with cancer cachexia hyperlipidemia is frequently observed. The authors suggested that lipolysis may be activated by inflammatory mediators and/or activation of beta-adrenergic receptors [4,36,41] and tumor-derived lipid mobilizing factors [36,43].

Wenjuan Di et al. studied the level of miR-146-5p in colon cancer patients and mouse model. The study included a group of 48 patients, and the obtained results were compared with the control group (healthy volunteers, *n* = 48). The authors observed increased expression of miR-146-5p, which resulted in lipolysis, fat mass loss, reduction and browning of WAT, as well as decreased tissue oxygenation [44]. Wu et al. demonstrated increased expression of miR-155 in cocultures of breast cancer cells and adipocytes. They also observed a correlation between this miRNA and WAT browning, increased adipocyte differentiation, catabolism, and lipolysis in adipose tissue. The authors found that miR-155 caused changes in adipocyte metabolism by reducing peroxisome proliferator-activated receptor *γ* (*PPARγ)* expression. PPARγ is a catabolism modulating marker and is involved in lipid accumulation [45]. A study by Kulyte et al., including 21 patients with gastrointestinal neoplasms with cachexia (*n* = 10) and without cachexia (*n* = 11), showed an increased level of miR-378 and decreased levels of miR-483-miR-5p, miR-23a, miR-744 and miR-99b. Kulyte et al. conclude that overexpression of miR-378 in human adipocytes induced enhanced lipolysis, which could play a key role in the process of fat loss in patients with cancer cachexia [46]. A level of miR-410-3p was measured in 60 individuals with gastric cancer in a study by Sun et al. One-half of the study group had cachexia. Authors observed that an increase in miR-410-3p expression inhibited adipogenesis, while a decrease in its expression resulted in the opposite effect. The authors showed that both the expression of miR-430-3p and its effect on adipogenesis are related to binding to the non-coding region of the 3’ Insulin Receptor Substrate 1 (3’IRS 1). Moreover, they observed that in the cachectic group, increased expression of miR-430-3p inhibited adipogenesis and lipid accumulation in adipose tissue. They also found that overexpression of miR-430-3p may be related to adipose tissue differentiation inhibition by lowering *IRS-1* gene expression [16] (Table 2). In patients with cancer cachexia, browning of adipose tissue, an increase in lipolysis and a decrease in the mass of adipose tissue were observed. Downregulated or upregulated different type of miRNAs can lead to changes in adipose tissue and the development of cancer cachexia.

#### 1.2.3. The Role of miRNAs in Cancer Cachexia-Related Muscle Tissue Remodelling

Due to their considerable degree of flexibility, skeletal muscles must be kept in good condition for lifelong health [47]. They play an important role in movement and maintaining body structure. Moreover, they are responsible for the synthesis of glycogen and are the important deposits of amino acids. For the proper functioning of muscles, activation of the processes that regulate growth, development, metabolism, and regeneration is required. Muscle deterioration and atrophy can be caused by a variety of factors, including aging, chronic illnesses, and inactivity [47]. Loss of muscle mass is most often associated with wasting syndrome, in which an involuntary loss of skeletal muscle mass occurs. Numerous studies have revealed, that various factors may be involved in muscle atrophy observed in cancer patients. Inflammation and metabolic disorders caused by procachectic, inflammatory cytokines (TNF-α, IL-1, IL-6 and IFN-γ) released by the host in response to the ongoing neoplastic process or by the tumor itself are considered the main causes of cachexia-related muscle atrophy. Inhibition of myoblast differentiation, an increase in autophagy, or issues with the renin-angiotensin system, that may cause muscle deterioration is also enumerated among potential causes of this syndrome [48,49]. Pro-inflammatory cytokines like TNF-α impact muscle proteolysis by stimulating nuclear factor kappa-light-chain-enhancer of activated B cells (NF-κB) pathways and the ubiquitin-proteosome system, which are crucial for protein breakdown during atrophy [50].

He et al. studied progenitor muscle cells in a group of cachectic Lewis lung carcinoma (LLC) mice and wild-type C57B6 male mice. Authors found a higher number of apoptotic muscle cells in cachectic LLC mice than in tumor-free mice. Their results indicate that cachexia causes apoptosis of muscle cells. They additionally pointed out that microvesicles (MV) harbouring miRNAs produced from pancreatic and lung cancer cell lines may cause muscle cell apoptosis. He et al.found that miRNA-21 released from MV can induce myoblast apoptosis via activation of toll-like receptor 7 (TLR7) through the c-Jun N-terminal kinase (JNK) pathway [51]. Miao et al., in experiments performed in vitro and *in vivo*, showed how C26 tumour exosomes might affect C2C12 mouse myoblasts cell atrophy in cancer cachexia. Study group included 21 patients with colon cancer (without cachexia *n* = 13, with cachexia *n* = 8), healthy volunteers and male BALB/c mice without tumour (*n* = 8) and with C26 (*n* = 8). They demonstrated that miR-195a-5p and miR-125b-1-3p, which may be found in C26 exosomes, can induce muscle cell apoptosis leading to muscle wasting in colon cancer cachexia through the mediation of the *Bcl-2* gene [52]. Soares et al. conducted a study on 7-week-old BALB/c mice with colorectal cancer (C26 cells) implanted in the back. They showed that selected miRNAs can participate in the modulation of muscle atrophy by influencing the synthesis and function of mitochondria. The authors noted that miRNA-206 and miRNA-21 can act synergistically and decrease protein synthesis. This applies especially to miRNA-206, which targets the JunD, Smad1, Runx1, and Rheb and through them exerts an effect on muscle tissue. Soares et al.noted that the miRNA profile may differ in distinct catabolic states and that changes in gene transcripts and miRNA expression do not occur at the same time. The peak of changes in the gene transcripts related to muscle atrophy 3 days after their differentiation was observed, while the change in miRNA expression was observed 7 days after. Amongs to the studied miRNAs, miRNA-206 and miRNA-21 showed to have the highest potential of being inducted by muscle atrophy. Moreover, the authors concluded that the modulation of miRNA could be a promising target for therapy [53]. Lee et al. conducted a study on 8 LLC C57BL6/J mice and 6 control C57BL6/J mice. The authors attempted to determine the miRNA profile responsible for the atrophy of muscle tissue induced by cancer cachexia. They identified 9 miRNAs that showed significant differences in expression. The authors divided miRNAs into functional groups, including those responsible for cell development and signalling between cells. The analyzed gene network was revealed to be involved in the regulation of muscle mass (e.g., Akt, FOXO3). Authors suggest that selected miRNAs may become a novel therapeutic target in the treatment of cancer cachexia [54]. Narasimhan et al. tested the miRNA profile in a group of 42 patients with pancreatic and colorectal cancer (with liver metastasis) (*n* = 22 patients with cachexia; *n* = 20 patients without cachexia). They found a higher expression of 8 miRNAs: miR-199a-3p, miR-423-3p, miR-532-5p, miR-345-5p, miR-1296-5p, let-7d-3p, miR-423-5p, miR-3184-3p. In addition, the authors reported which genes are regulated by upregulated miRNAs. As revealed, they may play a role in the innate immune response and inflammation (RPS6KA6), myogenesis and adipogenesis (DLK1, BMPR1B, SULF1), and the signal transduction pathways (DKK2, SFRP4). The authors suggested that all the above-mentioned genes may have a direct or indirect influence on the development of cachexia in patients [55]. In their study, Van de Worp et al. included 26 patients with NSCLC (non-cachexia *n* = 11 and cachexia *n* = 15) and 22 healthy controls. Among the genes functionally related to the studied miRNAs, they identified 22 pathways related to the regeneration or degradation of muscle cells, which include e.g., pro-inflammatory (Il-6, TNF-α), PI3K–Akt, and insulin pathways. Van de Worp et al.found differences in miRNA expression: miR-424-5p, miR-424-3p, miR-450a-5p, miR-144-5p, and miR-451a between NSCLC patients with cachexia and healthy volunteers. Moreover, they showed significantly higher expression of miR-424-3p in NSCLC patients without cachexia compared to healthy controls. The authors suggested that the assessment of the differences in the expression of miRNAs involved in the regulation of pathways related to the function of muscle cells in NSCLC patients with cachexia may facilitate the identification of potential therapeutic targets [56]. Chen et al. conducted a study on two types of breast cancer models (aggressive and benign, *n* = 79) and mouse model (MMTV-PyMT (*n* = 4) and MMTV-Her2 (*n* = 4) transgenic mice. Authors assessed circulating miRNA levels and their expression in the heart and skeletal muscles. They found that miRNA-486 expression was deregulated in tested tissues. Chen et al. demonstrated in vitro that TNF-α can induce a downregulation of miR-486 expression. Study shown that downregulated of miR-486 causes muscle atrophy, downregulation of *PTEN* and *FOXO1A* genes, MyoD transcription factor and decreased signalling through PI3K/Akt pathway [57]. In their study, Okugawa et al. analyzed 287 serum samples and matched surgical tissues (serum, *n* = 153; cancerous tissue, *n* = 134) from colorectal cancer (CRC) patients. They noted that in CRC patients with low psoas muscle mass index (PMI) the expression of miRNA-21 was significantly higher in the tissue as compared to the serum. Based on the results, the authors concluded that the assessment of miRNA-21 in the serum of CRC patients may facilitate decisions regarding nutritional treatment [58]. Pin et al. studied the profile of MV miRNA isolated from the conditioned medium collected from C26 or LLC cells, plasma of healthy BALB/c mice and Wistarrats, plasma-derived from C26 host or rats bearing the AH-130 hepatoma. A total of 118 miRNAs in MVs derived from the plasma of the C26 hosts were found. However, only three of them were down-regulated: miR-181a-5p, miR-375-3p, and miR-455-5p. No correlation among miRNAs in the MVs obtained from the blood of the C26 host and those released by C26 cells in the culture medium were observed. Moreover, the authors observed that overexpression of miR-148a-3p was associated with lower levels of Myogenic factor 5 (*Myf5)*, Myosin heavy chain 7 (*Myh7)*, and Myogenin (Myogenic factor 4- *Myog)*. In turn, overexpression of miR-181a-5p was accompanied by decreased levels of Myosin heavy chain (*MyHC)*, Myosin heavy chain 2 (*Myh4)*, and *Myh7*. The authors observed a lower level of *Myh7* in samples with overexpression of miR-21-5p. The above factors are responsible for the proper formation and differentiation of the structure of the muscle tissue, especially during the regeneration process. If their expression is reduced, it may additionally increase the degeneration process during cancer cachexia. The authors suggest that loss of muscle mass in C26-bearing mice was associated with disturbance in myogenesis. Moreover, they found that the lower expression of *MyHC* was accompanied by a decrease in protein synthesis [59,60]. Okugawa et al. performed a study including 183 patients with CRC. They assessed the level of miR-203 in cancer tissue and serum. The authors showed a negative correlation between the expression of miR-203 and the PMI level. They indicate that in CRC patients, higher miR-203 expression may inhibit cell proliferation and induce apoptosis of muscle cells by inhibiting surviving (target gene *BIRC5*) [61]. Gomes et al. investigated the effect of exercise on the expression of several miRNAs in serum and skeletal muscle in a mouse cancer model. Gomes et al. conducted a study on mice transfected with colon cancer (mice divided into two groups: non-exercising *n* = 11 and exercising *n* = 8) and BALB/c female healthy mice (*n* = 17) or with breast cancer (mice divided into two groups: non-exercising *n* = 7 and exercising *n* = 6) and c57 healthy female mice (*n =* 13). In both groups of mice with colon cancer higher expression of miRNA-486, lower expression of PI3K/mTOR pathway proteins, and higher expression of PTEN in the tibial muscle. Moreover, in both groups, muscle dysfunction, muscle wasting, and weight loss were observed. In the group of mice with breast cancer not exercising, decreased skeletal muscle function was observed. In the group of mice suffering from colon cancer, the downregulation of miRNA-486 and miRNA-206 was demonstrated. However, in the group of mice with breast cancer, the authors showed downregulation in miRNA-486 and upregulation in miRNA-206. The authors suggested that cancer reduces miRNA-486 expression, and exercise does not protect against changes associated with its lower level. However, in exercising mice with decreased expression of miRNA-206, muscle function was preserved. The authors believe that changes in miRNA-206 expression may be a potential biomarker of colon and breast cancer [62]. Xie et al. investigated the role of miR-29c and Leukemia Inhibitory Factor (LIF) in lung cancer cachexia using cell lines and a C57BL/6 mouse model transfected with LLC (*n* = 3). LIF is a member of the interleukin-6 family of cytokines. By regulating signal transduction mechanisms (JAK/STAT, MAPK and PI3K) it may positively or negatively affect the proliferation, differentiation and survival of various cells. Xie et al. showed that miR-29c expression was reduced and negatively correlated with muscle catabolic activity in mice with cancer cachexia. On the other hand, overexpression of miR-29c alleviated the symptoms of cachexia. The authors concluded that the miR-29c-LIF axis may become a therapeutic target in cancer cachexia [63,64] (Table 3). In the context of cancer cachexia, the impact of miRNA changes on muscle tissue is the most widely described. The cited studies are diverse, both in terms of miRNA and the study groups. However, it can be seen that downregulated or upregulated miRNAs can lead to changes in muscle tissue and the development of cancer cachexia. Figure 1 shows the influence of selected miRNAs on the immune system, adipose tissue and muscle tissue in the context of the development of cancer cachexia.

## 2. Summary

The available research suggests that miRNAs, due to their properties, e.g., the possibility of regulating from tens to hundreds of different genes, signalling pathways, and biological processes by one miRNA, but also their stability in biological material, the fact, that the change in their level reflects the disease status or the response to the applied treatment, they have great potential to be used as valuable biomarkers in the diagnosis, treatment, and prognosis of cancer cachexia.

## Figures and Tables

**Figure 1 jpm-12-01438-f001:**
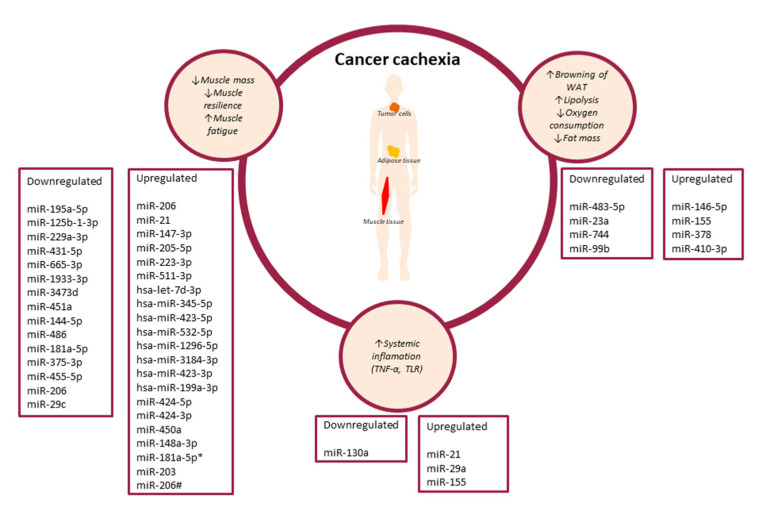
The influence of selected miRNAs on the immune system, adipose tissue and muscle tissue in the context of the development of cancer cachexia. (↓—decrease, ↑—increase).

**Table 1 jpm-12-01438-t001:** The role of miRNAs in the regulation of proinflammatory cytokines involved in cancer cachexia.

Authors (Year of Publication)	Study Group/Cancer Type	Race/Nationality	Anticancer Treatment (Surgery, CTH, RT)	Study Material	Studied Mirna	Time-Point of miRNA Assessment (before or after Anticancer Treatment/Experimental Model Manipulation)	miRNA Change in Patients with Cachexia or in Cachectic Models	miRNA-Related Targets/Changes in the Studied Material
Powrozek et al. (2018) [33]	Patients with HNC (*n =* 70, without cachexia, *n =* 33, with cachexia, *n =* 37)	Poland	IMRT	Plasma	miR-130a	Before	Downregulated	TNF-α
Fabbri et al. (2012) [34]	B6 mice with LLC (*n =* 18) Cell Cultures (A549, SK-MES, HEK-293)	-	-	Cell Culture/animal model	miR-21 miR-29a	-	Upregulated	TLR (murine TLR7, human TLR8)
Yehia et al. (2021) [35]	Patients with pancreatic cancer or NSCLC (*n =* 203, without cachexia, *n =* 94, with cachexia, *n =* 109)	Egypt	CTH (GEM+CIS)	Serum	miR-155	After	Upregulated	TNF-α (through TNF- α: SOCS1, TAB2, Foxp3)

**Table 2 jpm-12-01438-t002:** The role of miRNAs in cancer cachexia-related adipose tissue remodelling.

Authors (Year of Publication)	Study Group/Cancer Type	Race/Nationality	Anticancer Treatment (Surgery, CTH, RT)	Study Material	Studied miRNA	Time-Point of miRNA Assessment (before or after Anticancer Treatment)	miRNA Change	miRNA-Related Targets/Changes in the Studied Material
Wenjuan Di et al. (2021) [44]	Patients with colorectal cancer (*n* = 48) Healthy volunteers (*n =* 48) Male C57BL/6 mice (*n =* 6)	Asian (China)	-	Cancerous tissue samples	miR-146-5p	-	Upregulated	Browning of WAT Accelerated lipolysis Decreased oxygen consumption Fat mass loss
Wu et al. (2019) [45]	Patients with breast cancer (*n =* 108) Cell culture (C2C12)	-	-	Cell culture	miR-155	-	Upregulated	Browning of WAT Anomalous conversion and increased catabolism of muscle cells Lipolysis Muscle loss
Kulyté et al. (2014) [46]	Patients with gastrointestinal cancer (pancreas, stomach, liver metastasis) (*n =* 21, without cachexia, *n =* 11, with cachexia, *n =* 10)	Sweden	Surgery	Abdominal subcutaneous adipose tissue	miR-483–5p miR-23a miR-744 miR-99b miR-378	Before	Upregulated (miR-378) Downregulated miR-483-5p miR-23a miR-744 miR-99b	Accelerated lipolysis (miR-378)
Sun et al. (2021) [16]	Patients with gastric cancer (*n =* 60, without cachexia, *n =* 30, with cachexia, *n =* 30)	Asian (China)	Surgery	Exosomes from serum/tissue	miR-410-3p	Before	Upregulated	Adipogenesis inhibition

**Table 3 jpm-12-01438-t003:** The role of miRNAs in cancer cachexia-related muscle tissue remodelling.

Authors (Year of Publication)	Study Group	Race/Nationality	Anticancer Treatment (Surgery, CTH, RT)	Study Material	Studied miRNA	Time-Point of miRNA Assessment (before or after Anticancer Treatment)	miRNA Change	miRNA-Related Targets/Changes in the Studied Material
He et al. (2014) [51]	Wild-type C57B6 male mice with LLC (number of subjects not provided) Cell lines of lung cancer and pancreatic adenocarcinoma (A549, LCC, H460, AsPC-1, Panc-2, MEFs MDA-MB-231, MIA-PaCa, MCF7, C2C12)	-	-	Wild-type C57B6 male mice serum Cell lines	miR-21	-	Upregulated in myoblasts (cell line)	Myoblast apoptosis
Miao et al. (2021) [52]	Male BALB/c mice without tumour (*n* = 8) and with C26 (*n* = 8) Patients with colon cancer (*n* = 21, without cachexia *n* = 13, with cachexia *n* = 8) Healthy volunteers (*n* = 19) Cell line (C2C12)	Asian (China)	-	Serum and exosomes from mice and human subjects Mice muscle tissue Cell line	miR-195a-5p, miR-125b-1-3p	-	Upregulated	Downregulation of *Bcl-2* gene Muscle atrophy
Soares et al. (2014) [53]	BALB/c mice with colon carcinoma (C26) (*n* = 3) Cell line (C2C12)	-	-	Muscle tissue Cell line	miR-206 miR-21	-	Upregulated	Numerous different genes regulated by both miRNAs Muscle atrophy
Lee et al. (2017) [54]	C57BL6/J mice with LLC (*n* = 8) or without LLC (control, *n* = 8)	-	-	Muscle tissue	miR-147-3p miR-299a-3p miR -1933-3p miR-511-3p miR-3473d miR-233-3p miR-431-5p miR-665-3p miR-205-3p	-	Downregulated: miR-229a-3p miR-431-5p miR-665-3p miR-1933-3p miR-3473d Upregulated: miR-147-3p miR-205-5p miR-223-3p miR-511-3p	Muscle atrophy
Narasimhan et al. (2017) [55]	Patients with pancreatic and colorectal cancer (with liver metastasis) (*n* = 42, without cachexia *n* = 20, with cachexia *n* = 22)	Canada	Surgery	Muscle tissues (biopsies)	hsa-let-7d-3p hsa-miR-345-5p hsa-miR-423-5p hsa-miR-532-5p hsa-miR-1296-5p hsa-miR-3184-3p hsa-miR-423-3p hsa-miR-199a-3p	Before treatment	Upregulated	Numerous different genes regulated by studied miRNAs Muscle atrophy
van de Worp et al. (2020) [56]	NSCLC patients (*n* = 26, 11 without cachexia, 15 with cachexia) Healthy volunteers (*n* = 22)	Netherlands	-	Muscle tissue (biopsies)	hsa-miR-424-5p hsa-miR-451a hsa-miR-144-5p hsa-miR-424-3p hsa-miR-450a-5p	Before treatment	Upregulated: miR-424-5p miR-424-3p miR-450a Downregulated:miR-451a miR-144-5p	Muscle atrophy
Chen et al. (2014) [57]	Patients with breast cancer (*n* = 79) MMTV-PyMT (*n* = 4) and MMTV-Her2 (*n* = 4) transgenic mice Cell culture (C2C12)	USA	-	Human serumTransgenic mice models (serum, cancerous tissue, cardiac and skeletal muscle tissue) Cell culture	miR-486	-	Downregulated	Downregulation of *PTEN* and *FOXO1A* genes Decreased signalling through PI3K/Akt pathway Downregulation of MyoD transcription factor Muscle atrophy
Okugawa et al. (2018) [58]	Patients with colorectal cancer (serum, *n* = 153, cancerous tissue, *n* = 134)	Asian (Japan)	-	SerumCancerous tissue	miR-21	Before surgery	Upregulated	Sarcopenia Low PMI
Pin et al. (2022) [59]	BALB/c mice with or without C26 (*n* = 8) Male Wistar rats with or without AH-130 (*n* = 8) Cell lines (C2C12, LCC, C26)	-	-	Microvesicles from C26 or LLC cells Plasma of mice and rats Cell line	miR-181a-5p miR-375-3p miR-455-5p miR-148a-3p miR-181a-5p	-	Downregulated miR-181a-5p miR-375-3p miR-455-5p Upregulated miR-148a-3p miR-181a-5p	Upregulated miRNAs lead to downregulation of *Myf5*, *Myog*, *Myh4*, *Myh7* genes Delayed differentiation Disorganized mitochondrial system
Okugawa et al. (2019) [61]	Patients with colorectal cancer (*n* = 183) Cell line (SkMC)	Asian (Japan)	CTH (5-FU)	Cancerous tissue Serum Cell line	miR-203	Before surgery	Upregulated	Downregulation of BIRC5 Decreased skeletal muscles mass (decreased PMI) Skeletal muscle cells proliferation inhibition and apoptosis
Gomes et al. (2021) [62]	BALB/c female healthy mice (*n* = 17) or with colon cancer (*n* = 19) c57 healthy female mice (*n* = 13) or non-cachectic transgenic mice with spontaneous breast cancer (*n* = 16)	-	-	Serum Skeletal muscle tissue	miRNA-486 miRNA -206	-	Upregulated miR-206 (Breast cancer) Downregulated miR-206 (Colon cancer) miRNA-486	Muscle wasting
Xie et al. (2021) [63]	C57BL/6 mice with (*n* = 3) or without (*n* = 3) LLC Cell culture (C2C12)	-	-	C57BL/6 mice tissue Cell culture	miR-29c	-	Downregulated	Upregulation of LIF, JAK/STAT and p38 MAPK pathwaysMuscle wasting

## Data Availability

Not applicable.

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
