# Peer review of "Three Pathways of Cancer Cachexia: Inflammation, Changes in Adipose Tissue and Loss of Muscle Mass—The Role of miRNAs"

_jpm, 2022, doi:10.3390/jpm12091438_

Round 1

Reviewer 1 Report

This review by Homa-Mlak et al describes the role of microRNA and their potential use in the treatment of cancer cachexia. Remarkably, the authors present an extensive review of the current literature in vast areas. Given the complexity and the vast literature available, this is an excellent review.

Major comments:

- I suggest adding to the review a graphical scheme describing the major role of miRNAs in cachexia, and clearly distinguishing what are direct targets and downstream effects.

- "Despite that the number of studies reporting newly discovered miRNA still growing...". I think this statement should be corrected somehow, despite the fact that the number of miRNAs keep growing. Recent findings suggest that not all new entries are bonafide miRNAs (PMID: 34320405).

Author Response

This review by Homa-Mlak et al describes the role of microRNA and their potential use in the treatment of cancer cachexia. Remarkably, the authors present an extensive review of the current literature in vast areas. Given the complexity and the vast literature available, this is an excellent review.

Major comments: - I suggest adding to the review a graphical scheme describing the major role of miRNAs in cachexia and clearly distinguishing what are direct targets and downstream effects.

Response: Thank you for Your valuable suggestion. We have prepared and added a graphical scheme presenting correlation between different miRNA (including the direction of change in their expression) and alterations characteristic for cancer cachexia.

- "Despite that the number of studies reporting newly discovered miRNA still growing...". I think this statement should be corrected somehow, despite the fact that the number of miRNAs keep growing. Recent findings suggest that not all new entries are bonafidemiRNAs (PMID: 34320405).

Response: Thank you for Your valuable suggestion. We have corrected this statement according to the Reviewer's suggestion.

Reviewer 2 Report

The subject of this review is of great importance for the understanding of cachexia markers, for the search for new agents and therapeutic targets. And microRNAs may be the key to new biomarkers and understanding of the regulation of this syndrome.

I assess that this review work, despite recognizing the authors made an excellent bibliographic survey on the subject, should still undergo a more detailed review and improve the way of presenting ideas on this topic.

The works of the literature were exhaustively presented in each item on inflammation, alteration in adipose tissue or loss of muscle mass, all relating to the increase or decrease of microRNAs expression, but there was a lack of link between the presented ideas, explanations or conclusions by the authors. authors. The work is very descriptive but without any discussion by the authors to unify the various works found on microRNas as biomarkers.

Many sentences have been placed and have no connection with the following sentences, or any discussion and are often wrongly described, for example:

(lines 122-15) "They demonstrated that lung cancer cells release miR-21 and miR29a, which bind to immune cells' TLRs (murine TLR7, human TLR8) and trigger an inflammatory response. Tumor progression could also result from this mechanism. observed increased secretion of TNF-α and IL-6 by activating the NF-KB pathway [34]. What is TLR? How are microRNas linked to this structure? Are microRNAs considered Damage-associated molecular patterns (DAMPs)? "

Another examples:

-(lines 161-162) Wu et al. demonstrated increased expression of miR-in paraffin-embedded breast cancer samples from breast cancer patients (n=108) and C2C12 cell culture". Are there techniques that can detect microRNas expression in samples already prepared with paraffin? if yes, which technique was used , immunohistochemistry The expression of microRNAs are detected by qRT-PCR.

-(lines 165-166) "The authors found that miR-155 caused changes in adipocyte metabolism by reducing PPARγ expression. What is PPARy? What does it mean? No complementation by the authors.

(line 257) "They demonstrated in vitro that TNF-α can induce a change in miR-486 expression". TNF-α up- or downregulates miR-486, "change" is a general term.

(lines 272-274) Moreover, the authors observed that overexpression of miR-148a-3p was associated with lower levels of Myf5, Myh7, and Myog. In turn, overexpression of miR-181a-5p was accompanied by decreased levels of MyHC, Myh4, and Myh7." What do these factors mean? They are important transcription factors expressed during muscle tissue regeneration, in which no such importance was mentioned or discussed by the authors.

(line 299-301) "Xie et al. investigated the role of miR-29c and Leukemia Inhibitory Factor (LIF) in lung cancer using cell lines and a C57BL/6 mouse model transfected with LLC (=3)." LIF is an important myokine and its biological importance has not been mentioned.

The authors need to make a more concise conclusion on the whole exposed topic.

Others:

-Writing style- the authors exhaustively use "They" to mention the authors of the cited works;

-Some sentences are written incompletely, as an example on line 159.

-The references described by the authors' names at the beginning of the sentences are erroneously referenced, lacking the corresponding number of their reference.

-Too many words are joined together (may be text editing errors).

Author Response

The subject of this review is of great importance for the understanding of cachexia markers, for the search for new agents and therapeutic targets. And microRNAs may be the key to new biomarkers and understanding of the regulation of this syndrome.

I assess that this review work, despite recognizing the authors made an excellent bibliographic survey on the subject, should still undergo a more detailed review and improve the way of presenting ideas on this topic.

The works of the literature were exhaustively presented in each item on inflammation, alteration in adipose tissue or loss of muscle mass, all relating to the increase or decrease of microRNAs expression, but there was a lack of link between the presented ideas, explanations or conclusions by the authors. authors. The work is very descriptive but without any discussion by the authors to unify the various works found on microRNas as biomarkers.

Response: Thank you for Your valuable suggestion. We have added some new conclusions in the discussion as well as and new Figure , that unify the role of described miRNAs in the  development of cancer cachexia including their influence on the immune system, adipose tissue and muscle tissue.

Many sentences have been placed and have no connection with the following sentences, or any discussion and are often wrongly described, for example:

(lines 122-15) "They demonstrated that lung cancer cells release miR-21 and miR29a, which bind to immune cells' TLRs (murine TLR7, human TLR8) and trigger an inflammatory response. Tumor progression could also result from this mechanism. observed increased secretion of TNF-α and IL-6 by activating the NF-KB pathway [34]. What is TLR? How are microRNas linked to this structure? Are microRNAs considered Damage-associated molecular patterns (DAMPs)? "

Response: Thank you for Your valuable suggestion. We have revised this sentence as follows: “Fabbri et al. examined the level of several miRNAs in a B6 mouse model of lung cancer (n=18) and cell cultures (A549, SK-MES, HEK-293). They demonstrated that lung cancer cells release miR-21 and miR29a, which bind as ligands to immune cells' Toll-Like Receptors (TLRs) (murine TLR7, human TLR8) and trigger an inflammatory response. Tumor progression could result from this mechanism. They also observed increased secretion of TNF-α and IL-6 by activating the NF-KB pathway [34].”

TLRs can recognize death/damage-associated molecular patterns (DAMPs) and initiate the immune response. Interactions between TLR and miRNAs (which may be potential DAMPs) are still investigated. Kumar V, Barrett JE. Toll-Like Receptors (TLRs) in Health and Disease: An Overview. HandbExpPharmacol. 2022;276:1-21. doi: 10.1007/164_2021_568.

Another examples:

-(lines 161-162) Wu et al. demonstrated increased expression of miR-in paraffin-embedded breast cancer samples from breast cancer patients (n=108) and C2C12 cell culture". Are there techniques that can detect microRNas expression in samples already prepared with paraffin? if yes, which technique was used , immunohistochemistry The expression of microRNAs are detected by qRT-PCR.

Response: Thank you for Your valuable suggestion. We have revised this sentence as follows: “Wu et al. demonstrated increased expression of miR-155 in cocultures of breast cancer cells and adipocytes.”

-(lines 165-166) "The authors found that miR-155 caused changes in adipocyte metabolism by reducing PPARγ expression. What is PPARy? What does it mean? No complementation by the authors.

Response: Thank you for Your valuable suggestion. We have revised this sentence as follows: “The authors found that miR-155 caused changes in adipocyte metabolism by reducing peroxisome proliferator-activated receptor γ (PPARγ) expression. PPARγ is a catabolism modulating marker and is involved in lipid accumulation”

(line 257) "They demonstrated in vitro that TNF-α can induce a change in miR-486 expression". TNF-α up- or downregulates miR-486, "change" is a general term.

Response: Thank you for Your valuable suggestion. We have revised this sentence as follows: “Chen et al. demonstrated in vitro that TNF-α can induce a downregulation of miR-486 expression”

(lines 272-274) Moreover, the authors observed that overexpression of miR-148a-3p was associated with lower levels of Myf5, Myh7, and Myog. In turn, overexpression of miR-181a-5p was accompanied by decreased levels of MyHC, Myh4, and Myh7." What do these factors mean? They are important transcription factors expressed during muscle tissue regeneration, in which no such importance was mentioned or discussed by the authors.

Response: Thank you for Your valuable suggestion. We have revised this sentence as follows: ” Moreover, the authors observed that overexpression of miR-148a-3p was associated with lower levels of Myogenic factor 5 (Myf5), Myosin heavy chain 7 (Myh7), and Myogenin (Myogenic factor 4, Myog4).  In turn, overexpression of miR-181a-5p was accompanied by decreased levels of Myosin heavy chain (MyHC), Myosin heavy chain 2 (Myh4), and Myh7. The authors observed a lower level of Myh7 in samples with overexpression of miR-21-5p. The above factors are responsible for the proper formation and differentiation of the structure of the muscle tissue, especially during the regeneration process. If their expression is reduced, it may additionally increase the degeneration process during cancer cachexia. The authors suggest that loss of muscle mass in C26-bearing mice was associated with disturbance in myogenesis. Moreover, they found that the lower expression of MyHC was accompanied by a decrease in protein synthesis [59-60]”.

 (line 299-301) "Xie et al. investigated the role of miR-29c and Leukemia Inhibitory Factor (LIF) in lung cancer using cell lines and a C57BL/6 mouse model transfected with LLC (=3)." LIF is an important myokine and its biological importance has not been mentioned.

 Response: Thank you for Your valuable suggestion. We have added this sentence as follows: “LIF is a member of the interleukin-6 family of cytokines. By regulating signal transduction mechanisms (JAK/STAT, MAPK and PI3K) it may positively or negatively affect the proliferation, differentiation and survival of various cells.”

The authors need to make a more concise conclusion on the whole exposed topic.

 Response: Thank you for Your valuable suggestion. We have shortened the conclusions.

Others:

-Writing style- the authors exhaustively use "They" to mention the authors of the cited works;

Response: Thank you for Your valuable suggestion. We have improved the writing style throughout the manuscript.

-Some sentences are written incompletely, as an example on line 159.

Response: We have revised this sentence as follows: “The study included a group of 48 patients, and the obtained results were compared with the control group (healthy volunteers, n=48)”.

-The references described by the authors' names at the beginning of the sentences are erroneously referenced, lacking the corresponding number of their reference.

Response: We checked our manuscript and the references were corrected.

-Too many words are joined together (may be text editing errors).

Response: We checked our manuscript and the words joined together were separated.